# Position: Can Agentic AI Make Gig Economy More Fair?

## Abstract

The advent of Agentic Artificial Intelligence Systems (AAIS) is poised to revolutionize the gig economy by driving growth, expanding the workforce, and automating complex processes. AAIS achieves its performance by breaking down complex tasks into subtasks, leveraging multi-process frameworks to tackle intricate systems effectively. This approach often requires the use of multimodal or multi-language models, which are inherently susceptible to algorithmic biases. Furthermore, managing unstructured data—spanning natural language processing, images, videos, and meta-datasets—is indispensable for building robust AAIS. However, every aspect of unstructured data engineering is riddled with biases, which further amplifies the potential for unfair outcomes in the gig economy. Therefore, **We argue that deploying AAIS without addressing these systemic biases will inevitably compromise fairness in the gig economy.** To mitigate these challenges, we advocate for the urgent introduction of fairness assessment and mediation mechanisms tailored to AAIS, which are critical for fostering fairness in the gig economy.

## 1. Introduction

The deployment of Agentic Artificial Intelligence Systems (AAIS), which incorporate capabilities such as reflection, tool usage, planning, and multi-agent collaboration, dynamically adapting to changing environments without predefined behavior (Shavit et al., 2023), is becoming increasingly widespread (Durante et al., 2024). In a recent speech [1], renowned computer scientist Andrew Ng highlighted the growing trend of leveraging agentic AI-based large language models (LLMs) and large multi-modal models (LMMs) to create tools across various industries, emphasizing their potential to revolutionize workflows and drive innovation. The introduction of AAIS in the gig economy characterized by non-permanent, contract-based, or freelance workers performing on-demand tasks (Charlton, 2024)—seems inevitable, particularly for tasks such as job evaluation, task

---

[1] https://www.youtube.com/watch?v=KrRD7r7y7NY

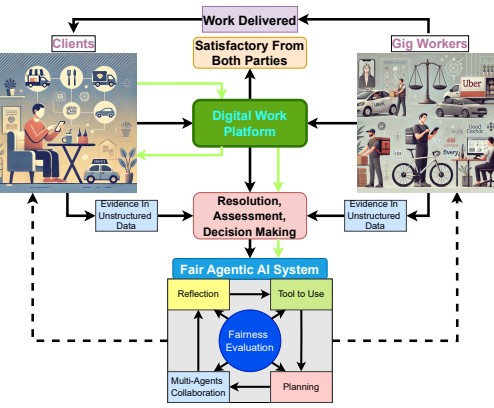

*Figure 1.* A **Fair** AAIS in the Gig Economy.

assignment, and replacing conventional human interventions. The gig economy can be broadly categorized into four main sectors: transportation-based services, handmade goods, household and miscellaneous services, professional services, and healthcare services. While this promises increased efficiency, scalability, and cost-effectiveness, it also raises critical concerns about fairness in the gig economy. AAIS autonomously makes decisions, contrasting with non-agentic workflows that depend on fixed policies(Durante et al., 2024). However, these systems face significant challenges in the gig economy due to inherent unfairness issues stemming from earlier model architectures and training paradigms. These biases, embedded in historical data or algorithmic frameworks, can perpetuate inequities in task assignments, performance evaluations, and compensation, posing obstacles to the fair and effective implementation of AAIS in the gig economy.

This issue is particularly urgent given the scale of the gig economy. According to a World Bank Group research report, there are an estimated 132.5 million full-time gig workers and 435 million online gig workers engaged in gig work as a secondary job, accounting for 4.4% to 12.5% of the global labor force (Datta et al., 2023). This underscores the pressing need to deploy fair AAIS. Our own research indicates that, in 2023, China had an estimated 84 million gig workers engaged in "new employment forms" (Daily, 2023). Similarly, McKinsey's 2024 American Opportunity Survey reported that 36% of employed participants—approximately

58 million Americans—are independent workers in the gig economy (Dua et al., 2022). In the European Union, the number of gig workers was 28.3 million as of 2022, a figure projected to reach 50 million by 2025 (Eur, 2024). Among gig workers across the US, UK, and Italy, data indicates that approximately 20% of gig workers in Italy and 10–15% in the UK rely on gig jobs as their sole source of income (Adermon & Hensvik, 2022). These statistics highlight the global significance of the gig workforce, not only in developing nations but also in developed countries. Furthermore, in many developed countries, gig workers are predominantly composed of low-skill youth, new immigrants with limited language proficiency, and migrants from refugee backgrounds (Adermon & Hensvik, 2022). Without robust fairness evaluations and policies, AAIS risks exacerbating disparities and disproportionately impacting vulnerable workers. Ensuring fairness in AAIS is therefore critical for fostering equity in the gig economy. Therefore, **we argue that ensuring fairness in AAIS before deployment is indispensable for achieving a fair gig economy.** Failing to address fairness in AAIS poses significant societal risks due to biases in decision-making processes. These biases primarily arise from three sources: dataset bias, algorithmic bias, and organizational bias in the implementation of AI models, all of which critically affect the components of AAIS. Dataset biases, for instance, are widespread in domains such as computer vision, natural language processing, video-language models, and voice systems (Mehrabi et al., 2021). Algorithmic bias often stems from issues related to interpretation, context transfer, algorithmic focus, processing, and training datasets (Danks & London, 2017). In this article, we focus on the impacts of the first two types of bias in AAIS within the gig economy. Notably, biases inherent in datasets can also exacerbate algorithmic bias, creating a feedback loop that reinforces unfair outcomes. Deploying AAIS in the gig economy without addressing these sector-specific biases risks flawed evaluations and assessments, ultimately resulting in biased decisions and inequitable outcomes. Addressing these biases is critical to ensuring fairness and preventing societal harm in AI-driven systems.

While some argue that ensuring fairness in AAIS could impose additional financial burdens on online gig platforms in gig economy, potentially exacerbating unfair pay, we believe such investments are indispensable for achieving long-term benefits in the gig economy. Fair AI systems can promote sustainable growth models for platforms while enhancing their reputations, attracting competent workers, and increasing market valuation. Moreover, while fairness evaluation and decision-making processes inevitably involve handling sensitive client and customer information, privacy concerns can be addressed with existing privacy-preserving methods, similar to those used in other applications.

To address fairness challenges in AAIS, we advocate for several measures. One key recommendation is to implement Fair AAIS Workflow, as well as establishing a precedent-based dataset for reference, ensuring long-term fairness across all sectors of the gig economy. Additionally, we propose adopting a Privacy-preserving mechanisms, evaluation metric tailored to the unique nature of customer-worker interactions and assessment mechanisms in each sector. This approach ensures sector-specific solutions that effectively address biases.

We also emphasize the importance of creating independent authorities responsible for auditing fairness in AAIS. Such bodies would play a crucial role in fostering transparency and ensuring the long-term sustainability of the gig economy. Furthermore, we recommend the deployment of long-term fairness evaluation metrics and Fairness Agentic Artificial Intelligence Technology Assessment (FAAITA) and fairness-focused assessments of agentic AI technologies before any online platform adopts these systems. Given the inherently interpersonal nature of interactions between clients and contractors, particularly in transportation and professional service sectors, we propose adapting a prisoner's dilemma-inspired framework within a instantaneous fair evaluation system to discourage parties from making biased and reduce potential face-to-face conflicts system. In addition, we explore the broader societal implications of deploying fair agentic AI-driven assessment and mediation systems across the gig economy. Transparent and fair foundation models play a critical role in preventing social conflicts, strengthening trust in the gig platforms, and protecting the rights of both workers and customers. By ensuring fairness and transparency, AAIS can foster equitable working conditions, promote collaboration, and enhance the overall equability of the gig economy.

## 2. Biases in AAIS

AAIS are autonomous decision-making systems with the potential to play an increasingly important role in the gig economy. These systems, which operate with a degree of autonomy and involve reflection, tool usage, planning, and multi-agent collaborations, are used in various applications, such as autonomous delivery robots, AI-powered scheduling systems, and fraud detection systems. However, AAIS face significant challenges related to biases, which can be introduced at multiple stages:

**Unstructured Data Collection:** Sampling methods can introduce bias, particularly when data is incomplete or reflects societal prejudices. For example, a dataset used to train a delivery robot might over-represent data from affluent neighborhoods, leading to poor performance in other areas.

**Algorithm Design:** The design of AI algorithms can introduce bias through the use of features that are proxies for

protected attributes, biased objective functions, or flawed assumptions. As a result, an AI-powered scheduling system that prioritizes workers with high ratings might inadvertently disadvantage newcomers or those facing discrimination. These biases can have significant consequences for gig workers, leading to reduced earning opportunities, reinforcing existing inequalities, and eroding trust in the system. (Deng et al., 2024) highlights how biases in datasets and algorithms contribute to unfairness in large language models and machine learning systems, ultimately resulting in unfair AAIS. In the context of the gig economy, this can have far-reaching implications for fairness and equity. Section 3 further explore the implications of these biases existed in AAIS to gig economy.

## 2.1. Bias in Unstructured Data

In order to achieve agentic artificial intelligence systems in the gig economy, dealing with unstructured data (images, textual inputs, voices, and utterances) is essential. However, bias is ingrained in every aspects unstructured data from language to computer vision.

### 2.1.1. BIAS IN TEXTUAL DATA

Bias in natural language dataset, arising from five primary sources—data selection, annotation, representations, models, and research design (Hovy & Prabhumoye, 2021)—has significant implications for fairness in AAIS. These biases not only influence NLP outputs but also exacerbate unfairness in decision-making processes when such systems are applied in the gig economy. Domínguez Hernández et al. (2024) highlights that textual data bias persists in foundation models, which are often trained on internet-scraped datasets known to encode harmful associations across protected categories such as religion, disability, gender, race, and ethnicity (Abid et al., 2021; Deshpande et al., 2023; Nadeem et al., 2020). These biases are further magnified at the intersections of these characteristics (Guo & Caliskan, 2021; Tan & Celis, 2019). When AAIS rely on biased NLP models for critical decisions—such as task assignment, performance evaluations, and dispute resolution in the gig economy—these biases can lead to systemic inequities. For example, biased language models may inadvertently prioritize or disadvantage workers based on gender, ethnicity, or other protected attributes, resulting in unfair task distributions or evaluations. Such outcomes not only harm individual workers but also undermine trust and fairness in the gig platforms. Addressing biases in NLP is therefore crucial to developing fair AAIS that ensures equitable opportunities, transparent decision-making, and sustainable growth in the gig economy.

### 2.1.2. BIAS IN VISION AND LANGUAGE DATA

Some commonly observed categories of dataset bias in Vision and Language models, along with their underlying causes, include **Unequal Representation**, **Stereotypical Associations**, and **Annotation Bias** (Fabbrizzi et al., 2022). Unequal representation occurs due to demographic biases (e.g., age, race, gender) or the disproportionate representation of certain concepts in training data. This results in models that either perform poorly for underrepresented groups or exhibit discriminatory patterns. For instance, Buolamwini & Gebru (2018) highlights significant discrepancies in gender classification tasks, where darker-skinned females, lighter-skinned females, darker-skinned males, and lighter-skinned males experience unequal performance. Stereotypical associations arise when training data contains biased connections between visual elements (e.g., scenes, objects) and textual descriptions (e.g., captions of actions or features). Such biases perpetuate harmful stereotypes, such as associating specific genders with certain professions. For example, Hashimoto & Tsuruoka (2017) demonstrates that models often link women with household chores and cooking, while associating men with sports and professional occupations.

Annotation bias is introduced when cultural differences or backgrounds among annotators influence the labeling process for vision and language data. As shown by Ribeiro et al. (2020), annotation bias in human action labeling for videos can result in models that reflect these biases, leading to skewed performance and biased decision-making.

Studies (Gichoya et al., 2022; Yang et al., 2024; Glocker et al., 2021) indicate that deep learning models can encode sensitive information, such as race, gender, and age, leading to significant performance disparities across demographic subgroups. This issue is particularly prevalent in image datasets, including medical datasets (Jones et al., 2023). This phenomenon, often referred to as "unfairness," poses significant challenges to achieving equitable outcomes, especially in critical fields such as healthcare delivery (Gao et al., 2024). Lei et al. (2022) identifies a strong "static appearance bias" in commonly used video-and-language datasets. This bias suggests that current datasets and evaluation metrics fail to account for the dynamic nature of videos, as models focus on single frames of objects and scenes rather than sequences of actions and events.

### 2.1.3. BIAS IN SPOKEN LANGUAGE DATASET

Spoken language technologies, particularly Automatic Speech Recognition (ASR), have become crucial for enhancing efficiency in the gig economy. ASR plays a vital role in various applications, including online dispute resolution, customer service, emergency response, and voice assistants (Feng et al., 2024). This technology holds significant potential for improving AAIS, which acts autonomously on behalf

| Gig Sector | Description/Platforms | Key Fairness Challenges | Possible Solutions |
|---|---|---|---|
| Transportation | Ride-sharing, food delivery, courier services (Uber, Grab, DiDi, MeiTuan, Ola, Bolt) | Algorithmic bias in matching, pricing, route optimization; Dataset bias in training data | Algorithmic audits, fairness constraints, transparency; Diverse, representative datasets |
| Professional Services | Freelance work, consulting (Upwork, Fiverr, Freelancer) | Algorithmic bias in job matching, pricing, skill evaluation; Fair compensation, job security, IP rights | Fair ranking algorithms, transparent rating systems, worker protection; Fair payment systems, skill-based matching, IP protection |
| Handmade Goods & Services | Handmade goods, household services (Etsy, AirTask) | Algorithmic bias in search ranking, customer matching; Dataset bias in product categorization | Fair ranking algorithms, transparent search criteria, customer education; Diverse, representative datasets |
| Healthcare | Telemedicine, online consultations (Dr MyHealth360, Good Doctor Online, PlushCare) | Algorithmic bias in diagnosis, treatment, patient prioritization; Dataset bias in patient data; Racial/gender bias, data privacy, quality of care | Algorithmic audits, diverse training data, human oversight; Bias mitigation, robust data security, quality assurance |

*Table 1.* Challenges and Solutions for Fair Agentic AI in Each Sector of the Gig Economy

of users. However, biases within voice or spoken language datasets (Yadav et al., 2024; Sekkat et al., 2024) can lead to unequal performance across different demographic groups, raising concerns about fairness in Agentic AI. For example, ASR systems may exhibit recognition bias between genders or favor younger speakers over older individuals (Sekkat et al., 2024). This disparity poses a significant challenge, especially for older gig workers who rely on these technologies. Addressing these biases is essential to ensure fair and equitable outcomes for all users in the gig economy.

### 2.2. Algorithmic Bias in Agentic AI

Even when trained on unbiased data, AI models can still exhibit bias due to inherent design choices (Chen et al., 2023). Ruggeri & Nozza (2023) demonstrate that many vision-language models, used in applications like image captioning and visual question answering, perpetuate harmful stereotypes. In the gig economy, this can be particularly problematic. For instance, an AI system that assigns tasks to gig workers might unfairly discriminate based on a worker's appearance due to biases in its image recognition capabilities. Similarly, a customer service chatbot that relies on vision-language models might misinterpret visual cues from certain customers, leading to negative experiences. Wu et al. (2024) further underscores this issue by revealing fairness issues in both open-source and closed-source vision-language models, particularly in their performance across different genders and skin tones.

Considering the usage of AAIS, systems interacting with humans may reinforce user biases or adapt in ways that reflect biased user behaviour, especially when they learn from user input (Bousetouane, 2025). Besides the bias that manifest in individual cases, Agentic AI also produce non-immediate biases that result in delayed harms, for example, in the study of online hiring systems (Sühr et al., 2021). Without being aware of the potentially sensitive attributes, Hashimoto et al. (2018) also claims that users receive suboptimal performance will become discouraged and more likely to stop using the systems, making the disparity amplified over time. Based on these findings, we will not solely attribute such biases to various design choices (Mehrabi et al., 2021), such as feature selection, the objective function used for training, and the model architecture itself, but also to the broader ecosystem in which we deploy AI systems. For instance, we consider bias factors spanning reasoning, planning, and communication in LLM-powered systems, as well as different time granularities to redefine algorithmic bias. These findings highlight the importance of understanding the different types of biases that can arise in AAIS. While responding to ongoing biases, anticipating harms (Chan et al., 2023) for algorithmic biases are essential for developing and deploying of new systems, which is particularly valuable for mitigating potential risks in the gig sectors.

## 3. Implications of Unfairness in the Gig Economy

In this section, we examine the ramifications of algorithmic and dataset biases in AAIS and how they can hinder fairness in the gig economy.

### 3.1. Income Inequality

Many online gig platforms in these sectors depend on algorithms to manage and control various tasks; For instance, an algorithm might prioritize certain workers over others based on biased internal metrics, leading to *income inequality* and unstable earnings among workers with comparable performance levels. A study on ride-share matching algorithms found that small changes in system parameters could significantly influence income distribution among drivers, thereby creating uncertain outcomes (Bok'anyi & Hann'ak, 2019).

### 3.2. Time-based Stress

Furthermore, algorithmic management and compensation frameworks can increase stress and workload for gig workers. When workers perceive these systems as opaque or unfair, they often experience heightened *time-based stress* and concerns about procedural justice (Semujanga & Parent-Rocheleau, 2024).

## 3.3. Gender Inequality

Unfair algorithms contribute significantly to gender inequality, as highlighted by a study on the online healthcare platform such as Spring Rain Doctor (SRD) (Chen, 2024). Analyzing data from 13,472 physicians active between March 26 and June 30, 2020, the study found that female physicians, who made up 38% of the sample, were underrepresented in search results, appearing in only 30% of the top 50 searches. Female physicians also faced fewer consultations, providing 14.2% fewer services and charging 8% less on average compared to their male counterparts in the same specialty. Even after accounting for variables such as education, professional title, experience, and availability, significant gender gaps remained: a 2.3% price gap and an 11% gap in monthly consultation services. These disparities translate into female physicians earning 13% less than their male colleagues, amounting to ¥83.6 less per month. This evidence highlights that while the gig economy provides flexibility, it is far from a solution for gender inequality. Instead, it reflects and often amplifies societal biases through algorithmic systems, perpetuating inequities rather than addressing them.

## 3.4. Wage Discrimination

Unfair AAIS can also lead to algorithmic wage discrimination, for example, hourly pay is determined by regularly adjusted formulas that draw on complex data, including location, individual behavior, and market supply and demand. (Glick) highlights an example from a former board member of the worker-led labor rights organization Rideshare Drivers United, who once received approximately 80% of each ride fare. However, after Uber and Lyft implemented opaque algorithms that incorporate data collected from workers to determine pay for each ride, drivers began receiving significantly lower payouts with little explanation. This practice may explain the inconsistent earnings for drivers completing identical rides in the same area at the same time. By surveilling workers and analyzing their behavior, these companies appear to have created a system that exploits desperation to suppress wages. The algorithms seem to identify the lowest amounts that drivers are willing to accept and normalize them, contributing to the steep decline in earnings reported by many drivers.

## 3.5. Unfairness and Inequity in Healthcare Services

With over 40% of the global population lacking adequate access to healthcare (World Health Organization, 2016), the integration of AAIS, which utilizes large language models (LLMs), offers a promising opportunity to enhance global health. However, deploying AAIS in teleconsultation and diagnostic support also presents significant risks, particularly due to biases inherent in current LLMs. AAIS has potential applications in healthcare, such as interfacing with electronic health record systems and providing diagnostic assistance. Despite this promise, substantial fairness challenges persist, raising concerns about the potential harm these systems could cause. AAIS in healthcare has demonstrated evidence of perpetuating racial and gender biases, which could adversely affect patient care. For instance, a study (Omiye et al., 2023) evaluating four prominent LLMs found that their responses often reinforced race-based medical misconceptions. These models, when repeatedly asked the same questions, produced inconsistent answers and occasionally propagated debunked, racist ideas, highlighting their unreliability in sensitive medical contexts. Gender bias presents similar concerns. For example, when generating medical scenarios for conditions like sarcoidosis, LLMs disproportionately associated the disease with Black female patients, doing so in 81% of cases, despite its relevance across various demographic groups (Zack et al., 2024). While the models correctly linked conditions like rheumatoid arthritis and multiple sclerosis to female patients, they often failed to represent broader demographic diversity. Moreover, Hispanic and Asian populations were underrepresented unless associated with stereotypical conditions like hepatitis B or tuberculosis, revealing significant limitations in model outputs. Additionally, AAIS are not immune to cognitive biases that affect human decision-making. A study (Schmidgall et al., 2024) identified seven key biases impacting LLM performance: self-diagnosis bias, recency bias, confirmation bias, frequency bias, cultural bias, status quo bias, and false consensus bias. These biases can significantly reduce diagnostic accuracy, with performance drops of 10% to 26% in the presence of biased prompts. For example, self-diagnosis bias may cause clinicians relying on LLMs to overvalue a patient's self-assessment, potentially skewing medical decisions and exacerbating errors. The findings underscore that while AAIS can improve healthcare access and support, unresolved biases and inconsistencies could undermine patient safety and equity. Overrepresentation or underrepresentation of certain groups distorts risk estimation and diagnostic focus, while cognitive biases reduce diagnostic accuracy. These challenges necessitate robust regulatory frameworks to ensure that LLMs contribute to equitable healthcare outcomes without perpetuating harmful stereotypes or inaccuracies.

## 4. Alternative Views

### 4.1. Does Fair Agentic AI Increase Financial Burden and Exacerbate Unfair Pay?

The deployment of fair agentic AI entails significant short-term costs due to substantial initial investments, including the collection of unstructured datasets (e.g., images, textual inputs, voices, and utterances), data cleaning, and the design

of effective algorithms. For instance, a common approach to bias mitigation is re-annotation; however, this process incurs considerable additional expenses (Kheya et al., 2024). According to (von Zahn et al.), achieving fairness in AI, as evidenced in e-commerce, has led to an **8% to 10%** increase in total financial costs. We use e-commerce as an example because it shares many similarities with the gig economy and provides the only available evidence on the financial implications of achieving fairness in AI.

However, in the long term, fair agentic AI offers substantial tangible and intangible benefits in fostering fair pay. First, fair agentic AI can significantly reduce the burden and costs associated with dispute resolution, which often requires human intervention. For example, it can save legal expenses by eliminating the need to outsource unresolved disputes to tribunals (Lee & Cui, 2024). Second, establishing a reputation as a fair, agentic AI-based platform offers numerous advantages, such as enhancing the company's image, attracting loyal customers, retaining employees, and reducing marketing expenses—all of which contribute to fair pay. Gatzert (2015) highlights that socially responsible companies attract more competent applicants and experience lower employee turnover. According to a World Bank survey (Datta et al., 2023), the top reasons firms or clients choose to hire workers from specific online gig platforms include brand recognition, quality of services, effective dispute resolution mechanisms, and trust in the platform. By incorporating fair agentic AI, companies can enhance customer trust, improve financial performance, and promote equitable pay practices. Finally, neglecting to deploy fair agentic AI could damage a company's reputation, erode employee confidence, and lose client trust, ultimately leading to declining revenue and exacerbating unfair pay. Overall, deploying fair agentic AI not only reduces workflow burdens and minimizes reliance on human intervention but also enhances the company's reputation, resulting in higher market valuation, increased revenue, and improved employee retention—key factors in promoting fair pay.

### 4.2. Privacy Concerns in Data Collection for Fair AAIS in the Healthcare Gig Economy

While the deployment of fair AAIS offers both tangible and intangible benefits to the gig economy, particularly in the healthcare sector, the process of large-scale dataset collection raises significant privacy concerns. Data collection is a vital step in enabling these systems to function effectively, but the risks of data breaches, misuse, or unauthorized access cannot be overlooked. These concerns are especially critical in the health sector, where datasets often contain confidential and sensitive patient information that must be rigorously protected. Any data leakage could have serious repercussions, potentially disrupting patients' daily lives and eroding trust in the system.

Ensuring fairness in AAIS for the healthcare gig economy presents additional challenges, particularly due to the diverse nature of patient data across hospitals. Hospitals often specialize in treating specific illnesses, meaning that evaluating doctor performance or providing accurate, fair diagnoses requires access to a wide range of datasets. However, this creates a fundamental tension: how can sensitive patient data be protected while allowing AAIS access to the information necessary for optimal performance? To address these challenges, existing machine learning methodologies, such as federated learning, can be leveraged. Federated learning enables multiple entities (referred to as clients) to collaboratively train a model while ensuring that data remains decentralized, reducing the risk of privacy breaches (Hu et al., 2024; Yang et al., 2019). Additionally, techniques such as data encryption, classification of sensitive information, and stringent scrutiny of data accessibility rights can mitigate privacy concerns. By implementing robust privacy-preserving mechanisms—such as user consent frameworks and real-time monitoring of data usage—healthcare providers can strike a balance between fairness and privacy. Moreover, the successful deployment of fair AAIS requires not only technical safeguards but also clear regulatory frameworks and transparency measures to build trust among all stakeholders. Without addressing these privacy concerns, the push for fairness in agentic AI within the healthcare gig economy risks unintended consequences, including patient mistrust, potential misuse of sensitive data, and ethical dilemmas related to data ownership and accessibility. While fair AAIS have the potential to revolutionize the healthcare gig economy, these systems must be designed with privacy as a foundational principle to ensure equitable and trustworthy outcomes.

## 5. What Can We Do?

We suggest holistic recommendations, which are recommendations applicable to all sectors, and specific recommendations, which are tailored to individual sectors due to their unique characteristics.

### 5.1. Implementation of Fair Agentic AI Workflow

The following workflow consists of three parts: steps 1-3 deal with data preprocessing to remove dataset bias, step 4 addresses algorithmic bias by carefully designing the system, and step 5 focuses on fair evaluation.

**1. Data Transformation.** Convert utterances and other audio inputs into textual representations using techniques like Automatic Speech Recognition (ASR) and Natural Language Processing (NLP). This standardization facilitates further analysis and comparison. Acknowledge and address the challenges associated with this conversion, such as dealing with accents, dialects, and noise, which can impact the

accuracy and fairness of the resulting textual data.

**2. Bias and Deceit Detection.** Analyze data to identify biases or deceit using a combination of techniques include 1. **Sentiment Analysis:** Evaluate the emotional tone of the text to detect potential biases or discriminatory attitudes. 2. **NLP-Based Detection:** Apply NLP techniques to recognize discriminatory language, stereotypes, and hate speech that may perpetuate harmful biases. 3. **Machine Learning Models:** Train supervised models using labeled examples to automatically detect potentially problematic content related to bias and deceit.

**3. Fair Feature Engineering.** Carefully select features that ensure fairness and prevent the introduction or amplification of biases during model training which include 1. avoiding features that are highly correlated with protected attributes. 2. using fairness-aware feature selection techniques to identify and mitigate potential biases. 3. ensuring that features are relevant and meaningful for the task at hand, thereby reducing the risk of unintended discrimination.

**4. Planning and Execution.** Develop a comprehensive implementation plan for fair AI, covering software development, system integration, and ethical considerations. This plan should include: 1. Ethical guidelines for responsible AI development. 2. Defined accountability mechanisms for AI decisions. 3. Ensuring transparency and explainability in AI system decision-making processes.

**5. Fairness Evaluation.** Assess the fairness of the system using fairness metrics: 1.**Demographic Parity:** Ensuring that outcomes are distributed equally across different demographic groups to prevent disproportionate impacts. 2. **Equalized Odds:** Ensuring that false positive and false negative rates are similar across groups, minimizing discriminatory outcomes. 3. **Predictive Rate Parity:** Ensuring similar accuracy rates across different groups to prevent disparities in performance based on protected attributes. Continuous monitoring and evaluation are necessary to ensure that the system remains fair over time and adapts to changing data distributions and societal contexts.

### 5.2. Long-Term fairness

Fairness evaluation in AAIS (Agentic Artificial Intelligence Systems) should no longer be static, as these systems are becoming increasingly dynamic. Evidence suggests that static fairness measures can hinder long-term fairness (Deng et al., 2024). We propose that fairness evaluation for AAIS in the gig economy should adopt a long-term perspective, focusing on the sustained impact of these systems. Since AAIS involves autonomous agents that interact with their environment over time, evaluating fairness requires considering long-term outcomes. Long-term fairness was first explored by (Deng et al., 2024), where the future returns of different groups in the gig economy are compared under specific fairness or agent policies (Plecko & Bareinboim, 2022). We believe that long-term fairness is particularly critical in sectors such as transportation and healthcare within the gig economy, as they employ the largest number of workers. Implementing fair AAIS in the gig economy with a focus on long-term evaluation can help protect gig workers' rights by ensuring equitable task assignments, rewards, evaluation and compensation.

### 5.3. Fairness Agentic Artificial Intelligence Technology Assessment (FAAITA)

We believe it is of great importance to establish independent authorities responsible for auditing fairness in AAIS. To this end, we propose the Fairness Agentic AI Technology Assessment (FAAITA) as a framework to ensure the fair and effective development and deployment of AAIS, specifically tailored to the gig economy's healthcare sector. Recognizing the unique challenges of this sector, we have developed a separate, detailed assessment sheet focused on AI-driven medical interventions, provided in the appendix. The primary goal of FAAITA is to equip policymakers, funders, healthcare professionals with the guidance to evaluate the benefits, limitations, and comparative value of agentic AI technologies to ensure fairness in the gig economy healthcare context.

### 5.4. Establishing Precedent Cases Database for Reference

Establishing a shared dataset with examples of fair and unfair outcomes can serve as a valuable reference point for all sectors. This dataset can be used to train AI models, develop fairness metrics, and guide decision-making. It's crucial that this dataset is diverse and representative of the various demographics and scenarios within the gig economy

### 5.5. Privacy-preserving mechanisms

AAIS need for data sharing and collaboration introduces the risk of data leakage, especially during the exchange of information between different agents within the healthcare gig economy. Therefore, robust privacy-preserving mechanisms are essential. We can adapting Federated Learning to allows AI models to be trained on decentralized datasets held by different hospitals without directly sharing the data. The models learn from each dataset locally and only exchange model updates, preserving patient privacy. We can use differential privacy by adding noise to the data before sharing it with the AI system, making it difficult to identify individual patients while still preserving the overall statistical properties of the dataset. Lastly, we can use secure Multi-

Party computation which allows multiple parties to jointly compute a function on their private inputs without revealing those inputs to each other. This could enable collaborative analysis of patient data across different hospitals without compromising privacy.

### 5.6. Adapting Game Theory to Assessment Policy in the Gig Economy

Without fairness, unethical behavior may be inadvertently encouraged, leading to negative consequences for the system as a whole. Some clients fabricate complaints to fraudulently obtain refunds or discounts. However, platforms often favor clients in these disputes to maintain customer satisfaction. Research indicates that the social structure of intermediary marketplaces influences the impartiality of dispute resolution (Greetje(Gretta Corporaal, 2024). In relational marketplaces, long-standing client relationships make it more difficult for intermediaries to remain neutral, leading to biased outcomes against gig workers. When combined with commission-based incentives, dispute handlers may side with clients to secure future business. This unfairness can damage a worker's reputation, reduce earnings, and cause financial losses for the platform—particularly when complaints are not effectively validated. From the perspective of the Prisoner's Dilemma, individuals often make decisions based on their own interests, which may provide short-term benefits but ultimately lead to a worse outcome for the collective system. By introducing a fair agent—an impartial judge capable of detecting misconduct—the system can promote ethical behavior, encouraging participants in the economy to act responsibly.

| Client \ Contractor | Client-Truthful | Client-False |
|---|---|---|
| Contractor-Truthful | Fair resolution; trust maintained. | Contractor gains unfairly; penalized. |
| Contractor-False | Client gains unfairly; penalized. | Both penalized for false info. |

*Table 3.* Prisoner's Dilemma Game Theory in Preventing False Statements Between Client and Contractor

### 5.7. Instantaneous Fair Evaluation System for Reducing Potential Face-to-Face Conflicts

In transportation-based services, contractors and clients often interact in person or have access to each other's personal addresses. This proximity poses a significant risk if either party behaves inappropriately, as conflicts can escalate into physical violence or verbal altercations. For example, in one incident, a valet driver was driving on behalf of a client who had consumed alcohol. A dispute arose between the two, with accusations exchanged, nearly escalating into a physical altercation. To address such risks, we are suggesting an automated system capable of immediately and impartially addressing complaints as soon as one party files a grievance.

By leveraging voice analysis, the algorithm would evaluate the situation, provide a fair decision, and enforce penalties or corrective actions promptly. This approach aims to de-escalate tensions and prevent conflicts from evolving into physical or verbal confrontations, thereby significantly reducing the potential for offences. Reflecting on the above scenario, it is clear that developing a fair algorithm capable of delivering unbiased judgments is essential for resolving such disputes. We believe this research represents a critical and impactful direction.

### 5.8. Policy Advice on Fairness in AAIS for health Care Service

Existing regulatory frameworks for traditional AI systems, which often involve rigorous testing and validation for specific medical applications, fall short of addressing the complexities of AAIS. These frameworks do not fully account for the unique risks of unfairness that arise from AAIS' broad applicability and general-purpose design. Without clear and standardized processes, biases in AAIS could lead to disparities in healthcare outcomes, such as unequal treatment recommendations or misrepresentation of patient demographics. This highlights the urgent need for updated guidelines that address not only safety and accuracy but also fairness, ensuring that AAIS deliver equitable healthcare solutions for all. To regulate AAIS fairly, policymakers must strike a balance between fostering innovation and safeguarding equity in healthcare. Overly restrictive regulations risk stifling progress that could improve access and efficiency, while insufficient oversight could perpetuate biases that disadvantage certain populations. Fairness must be a foundational principle in developing regulatory frameworks, alongside transparency and robust auditing processes. Collaboration among policymakers, developers, and healthcare professionals will be critical in creating standards that ensure LLMs deliver fair and equitable outcomes in all sectors of gig economy.

## 6. Conclusion

In this position paper, we are arguing that the current AAIS is biased by showing evidence such as dataset bias and algorithmic bias. The potential consequences of such a system in the gig economy are discussed. We also break down the social implications and ramifications of each sector of the gig economy. Lastly, we suggest some measurements to achieve fair AAIS in the gig economy.

## Impact Statement

This paper presents work whose goal is to advance the fairness in field of Machine Learning for gig economy. There are many potential societal consequences of our work, none which we feel must be specifically highlighted here.

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

## A. Challenges and Solution

| Gig Sector | Description | Key Challenges | Possible Solutions |
|---|---|---|---|
| **Transportation-based-services** | Services that include offering transportation, such as food delivery, ride-sharing, or courier services. (Uber, Grab, DiDi, Cao Cao, MeiTuan, Ola, Bolt) | - Fairness in Algorithm Design: Algorithms may unintentionally discriminate against certain groups, leading to unfair treatment. - Worker Exploitation: Platforms may exploit workers by setting low pay rates, long working hours, and poor working conditions. - Data Privacy and Security: Platforms may collect and use personal data without proper consent or security measures. | - Algorithmic Audits: Regularly audit algorithms to identify and mitigate biases. - Transparent Pay Structures: Clearly communicate pay structures and ensure fair compensation. - Improved Working Conditions: Implement policies to protect workers' rights and well-being. - Robust Data Privacy Measures: Implement strong data protection measures and obtain explicit consent. |
| **Professional-services** | Services that include providing specialized skills or expertise, such as freelance writing, graphic design, IT outsourcing, or consulting. (Upwork, Clouddevs, Fiverr, Freelancer) | - Fair Compensation: Platforms may not always fairly compensate workers for their work. - Job Security and Stability: Gig workers often lack job security and stability. - Intellectual Property Rights: Disputes over ownership of intellectual property can arise. | - Fair Payment Systems: Implement transparent and fair payment systems. - Skill-Based Matching: Match workers with appropriate jobs based on their skills and experience. - Strong IP Protection: Establish clear guidelines for intellectual property ownership and usage. |
| **Handmade-goods, household, miscellaneous-services** | Services that involve producing and merchandising handmade goods, or providing household and miscellaneous services like cleaning, repairs, or personal care. (Lyft, Etsy, AirTask) | - Quality Control: Ensuring consistent quality of services can be challenging. - Customer Satisfaction: Maintaining high levels of customer satisfaction can be difficult. - Fair Competition: Preventing unfair competition from larger businesses. | - Quality Standards: Set clear quality standards and enforce them. - Customer Feedback Mechanisms: Implement effective feedback mechanisms to improve service quality. - Support for Small Businesses: Provide support and resources to help small businesses compete. |
| **Health-Care Services** | Services that involve teleconsultations, Online Health Prescription Consulting Service. (MaNa Dr, MyHealth360 and Doctor Anywhere (Singapore), Good Doctor Online (China), Prescription Lifeline and Dr. B (USA), PlushCare (USA), JustAnswer (USA)) | - Racial and Gender Bias: Algorithms may perpetuate existing biases in healthcare. - Data Privacy and Security: Protecting sensitive health information is crucial. - Quality of Care: Ensuring the quality of care provided through online platforms. | - Algorithmic Bias Mitigation: Implement measures to mitigate algorithmic bias. - Robust Data Security: Implement strong data security measures to protect patient privacy. - Quality Assurance: Establish quality assurance standards for online healthcare services. |

*Table 4.* Sectors in the Gig Economy which requires Fair Agentic AI

*Table 5.* Dataset Bias in AAIS and its Implications for the Gig Economy

| Type of Bias | Implications for Gig Economy Sectors |
|---|---|
| **Unequal Representation** (e.g., demographics, concepts) | • **Transportation-based services:** Unfair allocation of ride-sharing requests or delivery assignments, disadvantaging workers from under-represented groups.

• **Professional services:** Unequal opportunities for individuals based on demographic background, affecting project acquisition and income.

• **Healthcare services:** Inaccurate diagnoses or treatment recommendations for certain demographic groups, exacerbating health disparities.

• **Handmade goods, household, and miscellaneous services:** Reduced visibility and success for artisans or service providers from marginalized communities. |
| **Stereotypical Associations** (e.g., gender and profession) | • **Transportation-based services:** Biased ratings or customer preferences based on gender associations with vehicle types or driving behaviors.

• **Professional services:** Limited project offerings based on perceived gender roles, restricting career options and earning potential.

• **Healthcare services:** Assumptions about patient needs or preferences based on demographic characteristics, affecting care quality.

• **Handmade goods, household, and miscellaneous services:** Influenced customer perceptions of product quality or authenticity based on provider demographics. |
| **Annotation Bias** (e.g., cultural differences of annotators) | • **Across all sectors:** Perpetuation of harmful stereotypes and discriminatory practices, impacting fair treatment and opportunities for workers and customers. |

*Table 6.* Impact of Algorithmic Bias on Gig Economy Sectors

| Impacts of Algorithmic Bias | Impact on Gig Economy Sectors |
| --- | --- |
| **Bias in Agentic AI due to bias in data** | • **All Sectors:** May perpetuate harmful stereotypes and discriminatory practices, impacting fair treatment and opportunities for workers and customers. |
| **Income Inequality** | • **Transportation:** Unfair prioritization of drivers, leading to income disparities among those with comparable performance.

• **Prof. Services:** Unequal project allocation and pricing, resulting in income inequality among freelancers.

• **Healthcare:** Biased compensation and patient allocation, creating income disparities among healthcare providers.

• **Handmade etc.:** Unequal promotion and pricing of products/services, leading to income inequality among providers. |
| **Time-based Stress** | • **Across all sectors:** Increased stress and workload due to opaque and unfair algorithms, leading to burnout and negative mental health outcomes. |
| **Gender Inequality** | • **Across all sectors:** Perpetuation of gender biases and stereotypes, leading to unequal pay, fewer opportunities, and unfair treatment for women. Specifically impacts platforms like Spring Rain Doctor where female physicians experience underrepresentation and income disparities. |
| **Algorithmic Wage Discrimination** | • **Across all sectors:** Discriminatory wage determination based on factors like gender, race, or location, perpetuating pay gaps and economic inequality. Evident in ride-sharing platforms where opaque algorithms lead to inconsistent and potentially discriminatory payouts. |

## B. Overview of Fair Agentic Artificial Intelligence Assessment Processes

Through Fair Agentic AI Assessment we perform a systematic evaluation for any newly proposed medical intervention, taking into considerations of social, economic, organizational and ethical issues around its implementation. This includes developing new cost-effectiveness models and analysing budget impact to generate locally relevant evidence. This information can be very suitable for governments, healthcare payers, hospital managers, clinicians as well as general public to ensure efficient and sustainable for health care systems.

*Table 7.* Fair Agentic AI Assessment Sheet for Medical Interventions

| Category | Criteria | Score (Max) |
|---|---|---|
| **1. Technical Evaluation (25)** | **Accuracy/Performance:**

• How accurate is the AI?

• Evaluation metrics and comparison to existing methods.

• Limitations/biases in training and evaluation data. | 10 |
| | **Explainability/Transparency:**

• Can the AI's decisions be understood and explained?

• Transparency regarding data and algorithms. | 5 |
| | **Safety and Reliability:**

• Potential risks and harms.

• Robustness to errors and attacks.

• Mechanisms for safe and reliable operation. | 5 |
| | **Data Privacy and Security:**

• Protection of patient data and compliance with regulations.

• Measures to prevent unauthorized access/use. | 5 |
| **2. Social Impact (25)** | **Accessibility and Equity:**

• Accessibility for all patients, regardless of background.

• Impact on health disparities.

• Strategies for equitable access and benefit distribution. | 10 |
| | **Psychological and Emotional Impact:**

• Effects on patient well-being, trust, and patient-provider relationship.

• Considerations for emotional impact of AI-driven decisions. | 5 |
| | **Social Acceptance and Trust:**

• Public perceptions and attitudes towards AI in healthcare.

• Building and maintaining trust in the AI. | 5 |
| | **Impact on Healthcare Workforce:**

• Effects on roles and responsibilities of healthcare professionals.

• Job creation/displacement.

• Training and support for healthcare professionals. | 5 |

*Table 8.* Fair Agentic AI Assessment Sheet for Medical Interventions

| Category | Criteria | Score (Max) |
|---|---|---|
| **3. Economic Evaluation (25)** | **Cost-Effectiveness:**

• Costs of development, implementation, and maintenance.

• Potential cost savings and benefits.

• Cost-effectiveness compared to existing interventions. | 10 |
| | **Budget Impact:**

• Impact on healthcare budgets at different levels.

• Financial models for long-term budget impact assessment. | 5 |
| | **Return on Investment (ROI):**

• Expected ROI for investors.

• Financial risks and uncertainties. | 5 |
| | **Economic Disparities:**

• Potential for creating/worsening economic disparities.

• Financial assistance programs or mitigation policies. | 5 |
| **4. Ethical Considerations (25)** | **Autonomy and Informed Consent:**

• Respect for patient autonomy and informed consent.

• Procedures for patients to understand and control data use. | 5 |
| | **Beneficence and Non-Maleficence:**

• Prioritizing patient well-being and avoiding harm.

• Safeguards against unintended consequences or misuse. | 5 |
| | **Justice and Fairness:**

• Fair treatment of all patients.

• Addressing potential biases in algorithms or data. | 5 |
| | **Responsibility and Accountability:**

• Responsibility for AI's actions and decisions.

• Ensuring accountability for errors or adverse events. | 5 |
| | **Transparency and Explainability:**

• Transparent and explainable decision-making processes.

• Open communication about capabilities and limitations. | 5 |

