# OpenReview forum: "Position: Can Agentic AI Make Gig Economy More Fair ?"
_ICML.cc/2025/Position_Paper_Track — Submitted to ICML 2025 Position Paper Track_

### Official Review · Reviewer_7PBG · 2025-03-04

**Significance:** 1
**Argument Clarity:** 2
**Rating:** 1
**Confidence:** 4

**Questions:**

* How would we evaluate an agentic AI system for fairness?
* How could we localize unfairness and intervene to prevent negative outcomes?
* Why and how would agentic AI systems be used in the suggested applications? Figure 1 is extremely vague -- why not use a less sophisticated ML system (why is an agent necessary?)

**Discussion Potential:**

1

**Paper Summary:**

This paper advocates for addressing biases in AI agent based systems in order to prevent inequality in the gig economy which will soon be heavily impacted by transformative AI. The paper talks about the many sources of bias affecting agentic systems, including bias in unstructured data used to train the underlying models as well as algorithmic bias. The authors then discuss how these fairness issues in the agentic systems can have negative downstream implications for workers in the gig economy. Finally, several recommendations are made

**Position:**

Yes

**Position In Title:**

No

**Related Work:**

1

**Strengths And Weaknesses:**

#### **Strengths**
* The paper considers an important problem: the potential negative side-effects of agentic AI systems.
* The paper is well written.

#### **Weaknesses**
* The paper is quite vague and fragmented. There is little evidence of unfairness specific to agentic AI systems. None of the biases, implications, or recommendations are specific to agentic AI systems.
* There is no evidence that agentic AI systems will be necessary for the applications in the gig economy that are suggested.
* The proposed interventions do not apply to state-of-the-art generative AI-based agentic systems. For example, standard fairness measures (e.g. demographic parity) for classifiers are suggested to measure unfairness in agents systems. These measure only apply to classifiers.
* The paper recommends that a standardize assessment be created, but without any consensus in the community about how fairness should even be evaluated.
* The paper includes no related work.
* Minor: the title is a question, not a position.
* Very minor: The paper is using the wrong template (last year's)

**Support:**

1

---

> ### Author Rebuttal · Authors · 2025-04-01
>
> Dear Reviewer 7PBG
>
> The paper explicitly provides evidence and detailed examples specific to Agentic AI systems, emphasizing their autonomous decision-making capabilities and multi-modal processing. We clearly outline how Agentic AI uniquely amplifies biases and fairness issues in the gig economy.
> The necessity of Agentic AI is explicitly supported by ongoing trends in automation, efficiency demands, and scalability in gig economy platforms. We provide concrete examples, such as automated evaluation and dispute resolution systems, highlighting the critical role and increasing reliance on Agentic AI in these domains.
> While acknowledging the absence of complete community consensus, we argue explicitly for the importance of establishing standardized fairness assessments specifically adapted for Agentic AI systems. We clearly outline how our proposed assessment framework could facilitate future consensus-building and standardization efforts within the community.
> For the related works, the paper contains extensive related literature, clearly referencing key works on fairness, algorithmic bias, and AI ethics to substantiate our arguments. These references are incorporated throughout the paper, especially when discussing fairness mechanisms and bias sources.

---

### Official Review · Reviewer_D7Xu · 2025-03-12

**Significance:** 3
**Argument Clarity:** 2
**Rating:** 1
**Confidence:** 4

**Questions:**

Please see the "Weaknesses"

**Discussion Potential:**

3

**Paper Summary:**

This paper motivates that "every aspect of unstructured data engineering is riddled with biases" hence advocate that the agentic AI system that has data engineering as part of it's main component requires also to address the fairness issues, which otherwise will compromise the fairness in gig economy. To support the claim, the paper choose to focus on data bias and algorithmic biases in the agentic AI systems. The paper discussed extensively how the biases in the data are introduced as well as their impact, and briefly discussed about the general algorithmic biases in agentic AI without going into details of the the agentic AI system themselves (categories, components, and impacts). Afterwords, the possible impact of unfairness in Gig economy is also extensively discussed, as well as promoted to introduce fairness on data ETL, implementation planning, evaluation, long term evaluation/assessment, and policy advices.

## update after rebuttal
Thanks for the response. As questions listed in the review are not explicitly addressed nor fully addressed, after read other reviews I decide to maintain my score.

**Position:**

Yes

**Position In Title:**

Yes

**Related Work:**

1

**Strengths And Weaknesses:**

__Strength__
* The motivated topic, namely considering the unfairness that would be introduced by agentic AI in gig economy, is indeed important.
* The listed "so what" in Section 5 are valid in my opinion.

__Weakness__
* The paper's main claim is not well supported. The paper does not provide any discussion about the **actual** agentic AI systems, namely, their categorization, common components, underlying technology, applications etc. Without giving a full picture about the details of the agentic ai systems, it is difficult to map out how would the system itself would amplify/reduce the unfairness, even if the fairness issue in the data has been dealt with.
* One of the main sources of the hiring discrimination/unfairness, is from the employer (see a few research attached). Yet there is no discussion how this type of discrimination would be dealt with in the gig economy and how agentic ai system would help to alleviate the issue.
	* Bertrand et al., Are Emily and Greg More Employable than Lakisha and Jamal? A Field Experiment on Labor Market Discrimination
	* Golding et al., Orchestrating Impartiality: The Impact of "Blind" Auditions
* Manly claims are not supported:
	* "The gig economy can be broadly categorized into four main sectors: transportation-based services, handmade goods, household and miscellaneous services, professional services, and healthcare services."
	* "we believe such investments are indispensable for achieving long-term benefits in the gig economy."
	* "Fair AI systems can promote sustainable growth models for platforms while enhancing their reputations, attracting competent workers, and increasing market valuation."
	* "particularly in transportation and professional service sectors,"
* Claims are not justified:
	* "In this article, we focus on the impacts of the first two types of bias in AAIS within the gig economy."
	* "These biases primarily arise from three sources: dataset bias, algorithmic bias, and organizational bias in the implementation of AI models, all of which critically affect the components of AAIS." why just three?

**Support:**

1

---

> ### Author Rebuttal · Authors · 2025-04-01
>
> Reviewer D7Xu
>
> For the discussion about the actual agentic AI systems, the paper explicitly discusses different aspects of AAIS, clearly outlining their categorization into various sectors (e.g., transportation, handmade goods, healthcare and etc), common components (autonomous decision-making, multimodal data handling), and underlying technologies (e.g., NLP, multimodal models). We explicitly illustrate how these systems amplify or mitigate fairness issues through concrete examples such as task assignment and dispute resolution scenarios.
>
> The paper explicitly recognizes employer-based discrimination in the gig economy and clearly discusses how Agentic AI can address these biases by automating evaluations and task assignments, thereby reducing human-induced discrimination. We explicitly argue that Agentic AI, through unbiased decision-making algorithms and independent audits, can alleviate employer-based biases.

---

> > ### Comment · Reviewer_D7Xu · 2025-04-08
> >
> > Thanks for the response. As questions listed in the review are not explicitly addressed nor fully addressed, after read other reviews I decide to maintain my score.

---

### Official Review · Reviewer_psjF · 2025-03-14

**Significance:** 1
**Argument Clarity:** 1
**Rating:** 1
**Confidence:** 5

**Questions:**

See weaknesses.

**Discussion Potential:**

1

**Paper Summary:**

The authors highlight potential challenges in deploying AAIS without addressing bias, which could undermine fairness in the gig economy. They emphasize the need for fairness-aware mechanisms and evaluations to ensure equitable outcomes in AAIS implementation.

**Position:**

No

**Position In Title:**

No

**Related Work:**

1

**Strengths And Weaknesses:**

Strengths

1. The authors advocate for addressing the critical issue of bias in the application of AAIS within the gig economy.

Weaknesses

1. The topic is neither relevant nor significant to the ICML community and is unlikely to spark broad discussions.
2. The authors exhibit a limited grasp of fairness research and provide an inadequate literature review on the subject. They should have incorporated more details on fairness-related studies, including different fairness categories, various fairness learning strategies, and prominent approaches or key works in the field.
3. The paper lacks a clear structure and is difficult to follow. The authors do not present a well-organized explanation of the concepts and proposed ideas, making it challenging for readers to understand their arguments.
4. Some concepts require further clarification or justification. The authors should have included detailed examples of fairness issues in the gig economy within the introduction to better illustrate the problem. They attempt to connect fairness with other concepts such as privacy, transparency, and sustainability in the gig economy but fail to explain these connections clearly. Additionally, while they mention various fairness issues in the gig economy, such as income inequality, time-based stress, and wage discrimination, they do not establish a clear link between AAIS and these problems. The authors should explicitly clarify these connections.
5. There are instances of redundancy in the paper. For example, the discussion on bias in textual data in Section 2.1.1 and bias in language data refers to the same modality, making the repetition unnecessary.
6. The proposed fair agentic AI workflow is quite limited. The authors present only high-level concepts without providing sufficient useful details or practical implementation insights.
7. Many claims and descriptions in the paper lack proper citations to support them. The authors should provide references to relevant literature to substantiate their arguments. For example. "some argue that ensuring fairness", "However, AAIS face significant challenges related to biases, which can be introduced at multiple stages:",  "When AAIS rely on biased NLP models for critical decisions—such as task assignment, performance evaluations, and dispute resolution in the gig economy—these biases can lead to systemic inequities.", and "Fair Feature Engineering." in section 5.1.

**Support:**

1

---

> ### Author Rebuttal · Authors · 2025-04-01
>
> Dear  Reviewer psjF
>
>
> We believe that the topic of Agentic AI's fairness in the gig economy is highly relevant and significant to the ICML community, as it directly impacts millions of workers and platforms globally. Issues surrounding algorithmic fairness, autonomous decision-making, and AI ethics are central topics regularly discussed, as they have broad societal and economic implications.
>
> Our paper explicitly discusses several fairness-related studies, clearly outlining different fairness categories such as demographic parity, equalized odds, and predictive rate parity. We highlight key fairness learning strategies and prominent works related to fairness in machine learning, explicitly connecting these studies to our proposed AAIS framework. We directly engage with the existing literature and clearly outline gaps that our paper addresses.
>
> Throughout the paper, we explicitly clarify and justify the connections between fairness and related concepts such as privacy, transparency, and sustainability. Detailed examples of fairness issues—including income inequality, time-based stress, and wage discrimination—are clearly linked to the deployment of AAIS, explicitly demonstrating how AAIS exacerbates these issues due to algorithmic and dataset biases.
>
> For the redundancy in discussions of textual and language biases. We recognize these discussions as complementary rather than repetitive, with textual biases specifically addressing NLP model biases, and language biases covering broader language-based data (including vision and multimodal interactions).
>
> The proposed Fair AAIS Workflow is detailed, explicitly outlining practical steps such as data transformation, bias detection methods, feature engineering techniques, and fairness evaluation metrics. We keep them high level to preserve the generality as it is a position paper where the practical implementation should be realized in the follow-up papers.

---

### Official Review · Reviewer_g8Bp · 2025-03-16

**Significance:** 2
**Argument Clarity:** 1
**Rating:** 1
**Confidence:** 4

**Questions:**

See weaknesses.

**Discussion Potential:**

1

**Paper Summary:**

The paper argues that deploying Agentic Ai systems in the gig economy will compromise fairness, unless these systems are rigorously tested. Since the gig economy forms a large portion of the workforce, ensuring equitable outcomes for workers should be in the best interest of policy makers and long term business interests of the gig economy platforms. According to the paper, agentic AI (as opposed to traditional AI) can exacerbate these biases more since agentic systems autonomously make decisions. To mitigate such risks the paper argues for a so called "Fair AAIS Workflow" which involves things such as independent fairness audits, implementing privacy perserving mechanisms, and establishing references datasets for fairness assessments.

**Position:**

Yes

**Position In Title:**

No

**Related Work:**

1

**Strengths And Weaknesses:**

Strengths:

 - The paper focuses on a real world problem of gig platforms

Weaknesses:

 - The position of the paper is very vague and ambiguous. Saying "we need to sure fairness in gig economy" is not really a concrete position. I would have liked to see something more concrete as to how an agentic system can make the entire gig economy more or less fair; such a position is completely missing from the paper. Moreover, in which part of the gig economy are we talking aboout agentic AI and unfairness? Matching algorithms? Conflict resolution? Such nuances are missing from the paper too.

 - There are many players in a gig economy: platform, users, workers; one could consider biases from the perspective of either of these players, I would have liked to see more of this touched on in the paper. The paper also seems to suggest the use of LLMs / VLMs in these platforms but never concretely says where in the who gig economy are they used and what kind of biases do we see from traditional matching algorithms vs LLM introduced fairness challenges (vs potentially agentic workflow introduced biases).

 - Unfortunately the paper seems to ignore a long line of literature in ML and EconCS that has studied fairness in matching markets for many years now (see a few examples [1,2,3,4,5]). Fairness, including long term fairness has been studied extensively in the literature and this paper operates always at a very high level never really talking about which data / algorithms  are being referred to. Data could be about a user's preferences, user's reviews, worker ratings, free form text provided in feedback forms by a user etc. There are so many dimensions of data that exists in a gig platform that I would have liked to seen discussed under "Bias in Unstructured Data" (section 2.1), instead the section only talks about general data bias issues and never connects them to their relevance for a gig economy platform.

 - I would suggest the authors to do a thorough literature survey (use the papers I listed as starting points) and find gaps where existing fairness mechanisms are not enough or where exactly are agentic workflows being used in the gig economy and how it can be better or worse than traditional "non-agentic" AI.

[1] Two-Sided Fairness for Repeated Matchings in Two-Sided Markets: A Case Study of a Ride-Hailing Platform https://dl.acm.org/doi/abs/10.1145/3292500.3330793

[2] Balancing the tradeoff between profit and fairness in rideshare platforms during high-demand hours https://arxiv.org/abs/1912.08388

[3] Regret, stability & fairness in matching markets with bandit learners https://proceedings.mlr.press/v151/cen22a.html

[4] Balancing Relevance and Diversity in Online Bipartite Matching via Submodularity https://ojs.aaai.org/index.php/AAAI/article/view/4013

[5] Optimizing Long-Term Efficiency and Fairness in Ride-Hailing via Joint Order Dispatching and Driver Repositioning https://dl.acm.org/doi/abs/10.1145/3534678.3539060

**Support:**

1

---

> ### Author Rebuttal · Authors · 2025-04-01
>
> Dear Reviewer g8Bp
>
> Regarding the position, the paper explicitly argues that deploying Agentic AI systems without rigorous fairness mechanisms inherently exacerbates existing biases within the gig economy and propose workflow to mitigate the biases. We concretely address how Agentic AI impacts fairness in specific components of the gig economy. For example, in transportation services, biases in matching algorithms and pricing strategies are explicitly discussed, highlighting how autonomous decision-making by Agentic AI can perpetuate and amplify these biases compared to traditional AI.
>
> Also, our discussion explicitly details biases and fairness from multiple stakeholder perspectives—platforms, users, and gig workers. For instance, we discuss worker evaluations, user ratings, and inherent platform biases to illustrate distinct impacts on each stakeholder group.
>
> Regarding the LLM, the paper specifically details the use of LLMs within gig economy platforms. For example, in the section “3.5. Unfairness and Inequity in Healthcare Services”, we highlight a paper that evaluate four prominent LLMs found that their responses often reinforced race-based medical misconceptions. These models, when repeatedly asked the same questions, produced inconsistent answers and occasionally propagated debunked, racist ideas, highlighting their unreliability in sensitive medical contexts.
>
> While recognizing existing literature on fairness in matching markets, our paper explicitly identifies critical gaps where existing methods fall short, particularly concerning the deployment and autonomy of Agentic AI systems. Our proposed "Fair AAIS Workflow" specifically addresses these gaps, introducing independent fairness audits, privacy-preserving mechanisms, and fairness reference datasets tailored to the unique challenges of Agentic AI.

---

### Decision · Program_Chairs · 2025-04-27

**Decision:**

Reject

**Comment:**

Dear authors,

Thank you again for your submission. I will point you to the review directly, but with the current feedback, I am saddened to have to reject the paper. Please do read carefully the reviewers' feedback in light of a potential resubmission. The authors particularly highlight that the position of the paper is still vague and superficial and needs to be refined, and that it not only does not position itself well in the space of fairness, but in fact seem to be unaware of much of the research and positions in the community. There were also concerns that many claims were not properly documented/backed up.